# A Multi-Domain Group-Based Intervention to Promote Physical Activity, Healthy Nutrition, and Psychological Wellbeing in Older People with Losses in Intrinsic Capacity: AMICOPE Development Study

**DOI:** 10.3390/ijerph18115979

**Published:** 2021-06-02

**Authors:** Sergi Blancafort Alias, César Cuevas-Lara, Nicolás Martínez-Velilla, Fabricio Zambom-Ferraresi, Maria Eugenia Soto, Neda Tavassoli, Céline Mathieu, Eva Heras Muxella, Pablo Garibaldi, Maria Anglada, Jordi Amblàs, Sebastià Santaeugènia, Joan Carles Contel, Àlex Domingo, Antoni Salvà Casanovas

**Affiliations:** 1Fundació Salut i Envelliment (Foundation on Health and Ageing)—UAB, Universitat Autònoma de Barcelona, 08041 Barcelona, Spain; alexandre.domingo@uab.cat (À.D.); antoni.salva@uab.cat (A.S.C.); 2Navarrabiomed, Geriatrics Department, Hospital Complex of Navarra (CHN)—Public University of Navarra (UPNA), Navarra Health Research Institute (IdisNa), 31008 Pamplona, Spain; cesar.cuevas.lara@navarra.es (C.C.-L.); nicolas.martinez.velilla@navarra.es (N.M.-V.); fabricio.zambom.ferraresi@navarra.es (F.Z.-F.); 3Equipe Régional Vieillissement et Prévention de la Dépendance, Gérontopôle, Centre Hospitalier Universitaire de Toulouse, 31300 Toulouse, France; soto-martin.me@chu-toulouse.fr (M.E.S.); tavassoli.n@chu-toulouse.fr (N.T.); mathieu.ce@chu-toulouse.fr (C.M.); 4Ageing and Health Department in the Andorran Healthcare System, Servei Andorrà d’Atenció Sanitaria, AD700 Escaldes-Engordany, Andorra; eheras@saas.ad (E.H.M.); pagaribaldi@saas.ad (P.G.); manglada@saas.ad (M.A.); 5Chronic Care Program, Department of Health, Generalitat de Catalunya, 08028 Barcelona, Spain; jamblas_ext@gencat.cat (J.A.); sebastia.santaeugenia@gencat.cat (S.S.); jccontel@gencat.cat (J.C.C.); 6Central Catalonia Chronicity Research Group (C3RG), Centre for Health and Social Care Research (CESS), University of Vic/Central University of Catalonia (UVIC-UCC), 08500 Vic, Spain

**Keywords:** ageing, methods, frailty, exercise, nutrition, psychosocial intervention, mental health, lifestyle, guidelines

## Abstract

The World Health Organization has developed the Integrated Care of Older People (ICOPE) strategy, a program based on the measurement of intrinsic capacity (IC) as “the composite of all physical and mental attributes on which an individual can draw”. Multicomponent interventions appear to be the most effective approach to enhance IC and to prevent frailty and disability since adapted physical activity is the preventive intervention that has shown the most evidence in the treatment of frailty and risk of falls. Our paper describes the development of a multi-domain group-based intervention addressed to older people living in the community, aimed at improving and/or maintaining intrinsic capacity by means of promoting physical activity, healthy nutrition, and psychological wellbeing in older people. The process of intervention development is described following the Guidance for reporting intervention development studies in health research (GUIDED). The result of this study is the AMICOPE intervention (Aptitude Multi-domain group-based intervention to improve and/or maintain IC in Older PEople) built upon the ICOPE framework and described following the Template for Intervention Description and Replication (TIDieR) guidelines. The intervention consists of 12 face-to-face sessions held weekly for 2.5 h over three months and facilitated by a pair of health and social care professionals. This study represents the first stage of the UK Medical Research Council framework for developing and evaluating a complex intervention. The next step should be carrying out a feasibility study for the AMICOPE intervention and, at a later stage, assessing the effectiveness in a randomized controlled trial.

## 1. Introduction

Although the incidence of severe disability has decreased in recent decades, especially in developed countries, the proportion of people with mild or moderate disability has increased [1]. In addition, more than half of elderly patients are affected simultaneously by multiple diseases and take, simultaneously, four or more prescribed medicines, increasing the risk of drug-related interaction causing functional decline and side effects [2,3]. However, health classifications and therapeutic recommendations with a single-disease approach have traditionally undervalued several health conditions, such as the so-called geriatric syndromes that negatively affect health and quality of life [4], amongst which frailty stands out.

Frailty is defined as an age-related medical syndrome, caused by multiple causes and contributors negatively affecting the homeostatic reserves of the individual. This vulnerability predisposes the person to a high risk of negative outcomes [5,6]. Among them, frail older people with the lowest income and educational level are the most vulnerable [7]. Although a theoretical definition of frailty is almost universally agreed upon, there is a lack of corresponding consensus about the wide range of instruments that are available for use in clinical practice.

This situation raises the need for a paradigm change in the approach to older adults’ health [8]. After the publication of the active ageing model [9], the World Health Organization (WHO) has recently supported the creation of a new model of care focused on the preservation of functional capacity. This has been shown to be a good predictor of morbidity and mortality in older people [10]. Within this framework, the WHO has developed the Integrated Care of Older People (ICOPE) strategy, a program based on the measurement of intrinsic capacity (IC) as “the composite of all physical and mental attributes on which an individual can draw” [11]. The ICOPE strategy establishes the following five steps: screening for declines in domains of IC (step 1), assessment of environmental, health, and social needs (step 2), development of a customized care process advised by a person-centered appraisal (step 3), patient referral and supervision of care process (step 4) and getting involved in communities and support caregivers (step 5). Step 3 implies an integrated plan to carry out interventions dealing with declines in several domains of intrinsic capacity, which should be contemplated and put together. Moreover, self-management and prioritization of health objectives in accordance with patients’ needs, expectations and preferences should be a transverse feature of such multi-domain interventions [12]. As for step 5, it emphasizes the importance for older people to receive information about available community-based resources, and the need for health assets in their neighborhood to be involved in supporting care, according to recent social prescribing approaches [13]. The identification of functions and capacities contributing to the definition of IC is structured into five domains: cognition, mobility, vitality (which addresses poor nutrition), psychological (which addresses anxiety and depressive symptoms) and sensorial domain (vision and hearing) [14]. These domains and health conditions associated with IC interact at several levels, and many of their contributing factors can be modified. However, strict lockdowns adopted by governments in the context of the COVID-19 pandemic have had a negative impact on several domains of IC [15].

Mobility is a critical issue for healthy ageing and preventing dependence on care. In fact, structured and adapted exercise is the preventive intervention that has shown the most evidence in the management of frailty and risk of falls [16,17]. Recent guidelines have recommended, as part of initial treatment of frailty, the inclusion of a multimodal physical activity program with a resistance-based training component, and social support to increase adherence [18]. Physical exercise is associated with a decrease in the risk of mortality, chronic disease, institutionalization, and cognitive and functional impairment [19]. Particularly, programs including strength, balance, flexibility, and aerobic exercise have reported the greatest outcomes [20,21,22,23,24,25,26,27,28,29,30,31,32,33,34,35].

Supervised physical exercise programs addressed to older adults have been shown to contribute to an improvement of physical parameters such as cardiorespiratory fitness, muscle strength, gait and balance, and to reduce the risk of falls [21,22,26]. Therefore they are an effective intervention to delay several geriatric syndromes [23] and result in beneficial outcomes related to psychosocial and cognitive aspects, thus reducing symptoms of anxiety and depression [20].

Current physical activity and physical exercise recommendations for ageing suggest accumulating a minimum of 150 min of moderate aerobic physical activity or 75 min of vigorous aerobic physical activity and varied multicomponent physical activities three or more days a week, to improve functional capacity and prevent falls [30]. A recent systematic review describing physical exercise programs for older adults in Latin America [36] showed that interventions were mainly based on therapeutic physical exercise with a duration of 2–6 months and a frequency of 2–3 times a week with sessions lasting 30–60 min. The components of physical fitness that were exercised the most were muscle strength and cardiorespiratory fitness.

The Multicomponent Exercise Program Vivifrail^©^ is based on a series of exercises that allow, depending on the level of functional capacity, to gain arm and leg muscle strength, improve flexibility, retrain balance and coordination to prevent falls, as well as increasing aerobic endurance. The program has been shown to be safe and effective in preventing cognitive decline in hospitalized pre-frail and frail older adults [21]. This type of intervention has also been proven as the most effective to delay disability, cognitive impairment and depression [37] as well as effective to reverse the functional decline associated with acute hospitalization in very old patients [21].

As for the vitality domain, most programs addressing frail older people also include a nutritional intervention, as it has been proven to increase the gains of physical exercise [38,39,40,41]. The ICOPE strategy considers offering dietary advice and highlights the importance of overcoming barriers to people’s nutritional health. It also takes into account the social aspects of dining, particularly for those living alone or socially isolated, including arranging assistance with preparation and provision of food, identifying specific seasonal and proximity foods, and advising on adequate portion size [42]. Concerning the psychological domain, literature shows how several structured approaches and therapies, such as Cognitive Behavioral Therapy (CBT), problem-solving, behavioral activation and life review, are susceptible to be modified into brief interventions to address anxiety and depressive symptoms [43,44]. Regarding the latter, another critical issue to be considered is strengthening social support and staying socially connected to tackle loneliness and social isolation [45]. Providing a list of local community services available to older people and encourage their use to increase their participation in identifying potential barriers to community engagement [13] is another good practice. Researchers have also reported the effect of self-management and behavior change strategies that may help older people to increase adherence and adopt healthier lifestyles [46,47,48].

Thus, there is a need to develop and assess community-based interventions to enhance intrinsic capacity and prevent frailty and disability. Among them, and based on available evidence, multi-component interventions appear to be the most effective [11]. With respect to the type of interventions, and despite the fact that evidence is spread across different health conditions or specific lifestyle behaviors [49], groups have shown better outcomes and are commonly-used for a wide range of interventions, including those which have a strong focus on social support and behavior change [50,51,52,53]. Particularly, a recent systematic review has demonstrated that physical activity programs can be effective for reducing or delaying frailty but only when they are delivered in groups [16]. Other researchers have argued that group membership has also had a significant effect in maintaining and promoting health [54]. Finally, the utilization of groups has been justified based on time and resource savings [55,56].

The purpose of this article is to describe the development of a multi-domain group-based intervention addressed to older people living in the community aimed to improve and/or maintain intrinsic capacity by means of promoting physical activity, healthy nutrition, and psychological wellbeing.

## 2. Materials and Methods

In this study, we use the “Guidance for reporting intervention development studies in health research” (GUIDED) checklist [57] to detail the development process of AMICOPE (Aptitude Multi-domain group-based intervention to improve and/or maintain Intrinsic Capacity in Older PEople). We used the GUIDED checklist because it provides consensus-based reporting guidance for intervention development studies. It can also potentially show greater transparency, enrich quality and consistency, and improve knowledge about intervention development research and practice.

The text below describes the process of intervention development in relation to the 14 items of the GUIDED checklist. The intervention itself is described according to the TIDieR (Template for intervention description and replication) guidelines [58]. The intervention was developed in the context of two different projects. On one hand, the APTITUDE (Agir pour la PrévenTIon Transpyrénéenne de la Dependance chez les seniors) project [59], and on the other, the model for the prevention of disability and the promotion of personal autonomy in Catalonia [60]. APTITUDE is a European project funded by POCTEFA 2014–2020, which is the acronym of the INTERREG V-A Spain-France-Andorra Program. The APTITUDE project involves 11 different territories from Occitania, Andorra, Navarra, and Catalonia in the cross-border area of the Pyrenees. The general objective of APTITUDE is to prevent the dependency on older people by creating a network to promote care, training, research, and innovation in the areas of public health and gerontology. The network was structured with local coordination reference persons (*n* = 10–15) and operational teams (*n* = 50–100) in each territory. The model for the prevention of disability and the promotion of personal autonomy is a joint initiative of the Department of Health and the Department of Labor, Social Affairs and Families of the Catalan Government. This project has already started in five pilot territories and should be progressively implemented over the next years throughout Catalonia.

The target population corresponds to older people with losses in mobility, nutritional and/or psychological domains of intrinsic capacity, and without cognitive decline, visual impairment or hearing loss, living in the community and recruited or referred from primary care and community settings. Participants have to be able to attend the intervention on their own. Hence, we focus on a phase previous to frailty with a lower prevalence of diseases and disability, and without losses in the sensorial domain.

Our approach was theory and evidence-based, as well as coherent with the framework of the Medical Research Council (MRC) for the development and evaluation of complex interventions [61]. This framework is appropriate to be applied in those interventions including a number of components interacting with each other, several recipients and outcomes, different skills needed by the facilitators and a certain level of tailoring. During the intervention development process, decisions were taken in accordance with evidence from similar interventions and results from our previous studies [21,22,23,24,25,26,27,28,29,30,31,32,33,34,35,36,37,62]. We also integrated the recommendations of a working group, the appropriateness of different frameworks used in implementation research, and evidence shown by few strategies which seemed to be effective. Figure 1 illustrates how evidence from different sources enlightened the intervention development process of the AMICOPE intervention.

The theoretical rationale driving the design and the development of this multi-domain intervention was the ICOPE program of the WHO [11]. Rather than creating an entirely new intervention, AMICOPE was developed by incorporating components adopted from existing interventions that have already shown evidence in increasing functional capacity and improving mental health in older people. Hence, the Vivifrail^©^ program [63] was used for the physical activity domain and some methodologies were used in the “Feeling Well” program [64] (e.g., mapping party, photo-elicitation, goal setting) were applied for the psychological wellbeing domain and to promote behavior change. An intervention guide for the facilitators set up a framework to lead group activities adhering to general basic rules detailed in a decalogue and a common structure whilst taking into account participants’ preferences, needs and expectations.

A first working group reviewed the general scope of the intervention and reached an agreement on the basic components of the multi-domain intervention. This working group (*n* = 20) included professionals from different fields (geriatrics, neuropsychology, nutrition, pharmacy, primary care, occupational therapy, physiotherapy, public health, social work, sociology of health, sports medicine and sports science research). After literature review and draft development, evolving versions of the intervention were presented in several workshops and meetings to incorporate feedback from different stakeholders and to define specific contents and procedures [65]. Stakeholders included health and social care professionals, public administration, civic organizations (older people’s associations, patients’ associations, women’s associations, non-profit private foundations) and entrepreneurs of the silver economy. Due to the iterative and complex nature of the development process, some aspects were discussed and some changes affected the scope of the intervention. Hence, and even though it was beyond the intervention’s initial aims, specific contents about cognitive stimulation and medication review were suggested by the working group and finally incorporated as part of the final resulting intervention. Other important issues were discussed, including the setting and length of the intervention, and the duration of each of the sessions. There was a lot of consensus on the type of intervention (group vs. individual) and about who should deliver the intervention (the same facilitators for the whole intervention vs. a different expert for each session). At the end of the development process, some uncertainties remained mainly related to the implementation of the intervention in isolated, sparsely, and low-density rural areas of the Pyrenees, and the impact of the COVID-19 pandemic in the realization of group-based activities with older people.

The study follows the Code of Good Practice in Research (CBPR) adopted by the Universitat Autònoma de Barcelona (UAB). The current publication is open access and materials of AMICOPE multi-domain intervention are available on demand.

## 3. Results

The result of the study is a group-based multi-domain complex intervention described according to the TIDieR (Template for intervention description and replication) guidelines. The TIDieR checklist has been incorporated as Appendix A.

The intervention is called “AMICOPE” (Aptitude Multi-domain intervention to promote Intrinsic Capacity in Older PEople), and it is aimed at promoting physical activity, healthy nutrition and psychological wellbeing in older people living in the community. The guiding principle of our work was the ICOPE strategy and the conceptual framework of the intervention is described in Figure 2.

The materials to be used in the intervention consist in a detailed guide for the facilitators (Version 1.0, ©Fundació Salut i Envelliment UAB, Barcelona, Spain) that will be available after refinement under a Creative Commons 4.0 License, CC BY., and additional resources such as maps, photographs, and audio files with free intellectual property rights used to perform some other activities. The intervention procedures for physical activity will be based on the Vivifrail^©^ multi-component training program. It is based on a series of exercises that allow, depending on the level of functional capacity (i.e., severe limitation, moderate limitation and slight limitation), the development of arm and leg muscle strengthening and powering, balance retraining and coordination to prevent falls, and flexibility to improve heart health by walking. All the exercises are outlined in the procedure; guidelines for starting, frequency and progression to correctly monitor the instructions prescribed to the participant. To individualize the exercise program, initial functional capacity and risk of falls are assessed. Different functional capacity levels are determined based on the scores obtained from the Short Physical Performance Battery Test and the 6-m gait velocity test, with each leading to the recommendation of a certain customized multicomponent physical exercise program (Program A, B, B+, C, C+ or D).

Several group dynamics will be performed to promote social support and the exchange of personal experiences among participants, as well as the acquisition of self-management skills. Goal setting will be used to promote behavioral changes in the daily life of participants that are meaningful for them and that positively affect their healthy nutrition and psychological wellbeing. A pair of health and social care professionals with different backgrounds (nurse, physiotherapist, occupational therapist, nutritionist, psychologist, physical activity trainer, etc.) will be previously trained as group facilitators by the research team during a 30-h training program. A pharmacist will participate in the session about medication to support facilitators and to provide specific counseling to older people. The intervention will consist of 12 face-to-face sessions facilitated in groups of 8 to 12 older people. Sessions will be held weekly for 2.5 h over three months. Each session will include one hour of physical exercise using the Vivifrail^©^ (Pamplona, Spain) program and 1.5 h dedicated to any other intervention components. The intervention will be delivered in community facilities such as senior leisure centers, civic centers, or primary care centers, and in different locations of the surroundings. Particularly, 10 of the 12 sessions will take place in a space large enough to perform physical activity. For one session, the whole group will move to a grocery store to learn about nutritional facts. The remaining session will be devoted to visiting another senior center with the purpose of learning about programs and activities addressed to the community. The physical activity domain of AMICOPE includes individual prescription passports for participants tailored to their individual functional capacity, which will be assessed by the Short Physical Performance Battery, a walking speed test, and the risk of falls. The passports include instructions to perform exercises at home between sessions for two or more days a week. Nevertheless, other activities are intended to facilitate adherence to the intervention, enhance social cohesion and change lifestyles. Hence, participants bring personal objects to the sessions to introduce themselves and share SMART goals in accordance with their own preferences to increase self-efficacy. Finally, outings are chosen and agreed after a group mapping activity of local community assets and this allows each intervention to be—beyond individual interests and preferences—slightly adapted to a specific context. The intervention will be monitored by facilitators (or external observers during the pilot) with quantitative and qualitative indicators of adherence. Facilitators will get in contact with those participants not attending a session for the purpose of screening losses and to motivate them to attend the following week.

## 4. Discussion

In this study, we use the GUIDED checklist to describe the intervention development process of AMICOPE. We rationalize the process by detailing the context, the aim, the recipients, the theory and evidence-base findings, the utilization of previous experiences, the guiding principles, the participation of stakeholders, the changes made throughout the process, and remaining uncertainties. The resulting intervention is described using the TIDieR checklist. The study will help investigators and health professionals to design future complex interventions focused on the promotion of intrinsic capacity in older people and will provide better knowledge about intervention development research and practice.

The AMICOPE intervention is addressed to older people living in the community and aims to improve and/or maintain IC by promoting physical activity, healthy nutrition, and psychological well-being. A multimodal exercise program tailored to individual capacities and needs has been suggested as the most important approach to improving or maintaining locomotor capacity [11,20,21,22,23,24,25,26,27,28,29,30,31,32,33,34,35]. The significant contribution of nutrition to frailty has also been underlined by some authors [38,40]. However, other researchers have suggested that nutritional interventions delivered alone may not be effective for the management of frailty in older people [66]. As for the psychological domain, previous studies showed that group-based interventions addressing loneliness and social isolation could help to reduce depressive symptoms [45,62]. During the development process, specific and brief contents about cognitive stimulation and medication review were finally incorporated as part of the intervention, though it was beyond the initial aims of the intervention. This decision was taken on the basis that preventive cognitive training has benefits for older people, as reported by researchers [67]. A medication review can reduce polypharmacy by eliminating unnecessary, ineffective medications and those with a duplicative effect [68]. Nevertheless, scientific literature has evidenced that most factors related to losses in intrinsic capacity share the same underpinning physiological and behavioral causes [11]. Hence, interventions have benefits across different domains of IC. Physical exercise prevents loss of mobility but also has indirect preventive effects against psychological distress and cognitive decline [20]. Nutrition reinforces the effects of exercise [38,39,40]. Loneliness increases the risk of malnutrition [69]. Finally, we included goal setting as a cross-sectional element in our intervention, which has been considered by some authors as an effective behavior change technique as well as a fundamental component of successful interventions [70], including those promoting good dietary habits and physical activity [71] and, particularly, when integrated in complex interventions addressed to older people [72].

To date, the few examples found using the GUIDED checklist include two peer-reviewed articles reporting interventions aimed at improving mental health help-seeking behaviors for male students [73] and anxiety and depression management in patients with chronic obstructive pulmonary disease (COPD) [74], two pre-prints reporting interventions aimed to improve early diagnosis of cancer in primary care [75] and targeting antipsychotic prescription to nursing home residents with dementia [76], and one doctoral thesis reporting an arts-based intervention for patients with kidney disease [77]. However, some researchers have used the Consensus on Exercise Reporting Template (CERT) [78] to characterize physical activity programs for older adults in Latin America [36] and assess the feasibility of an enhanced prescribed exercise program in older acute medical patients [79] and the effectiveness of falls prevention interventions on reducing falls in hospitalized adults [80].

### Strengths and Limitations

To our knowledge, AMICOPE is the first group-based complex intervention aimed at improving some components of intrinsic capacity in older people, built upon the ICOPE framework. The use of the GUIDED checklist provides detailed information about the intervention development process and allow researchers to understand important aspects when developing multi-domain intervention addressing frailty and/or losses in IC. We hope this will help the scaling-up, replication, adaptation, or more comprehensive implementation of AMICOPE in other settings. We also think that the approach described in this study can be used as a model for future research in the development of complex interventions.

Despite these strengths, this study has several limitations. Although the intervention proposed in this study covers most of the IC components, we were not able to incorporate any components addressing visual deficiency and hearing loss, both related to the sensory domain. Otherwise, scientific literature is scarce for published intervention development studies and there are few examples in the literature using the GUIDED checklist. This could be because GUIDED has been published recently, in 2020, and may have not yet been integrated enough by researchers.

This study represents the first of four key stages of the UK Medical Research Council framework which consists of development, feasibility and piloting, evaluation, and implementation of a complex intervention. Once the AMICOPE intervention has been developed, its acceptability and feasibility should be assessed in a further step. Actually, the intervention was planned to be piloted in Catalunya and ten health and social professionals (six nurses, two physiotherapists and two physical activity trainers) were trained during November and December of 2019. A pilot of AMICOPE intervention addressed to a group of 12 older people started in Tortosa (Baix Ebre/Spain) in the beginning of February 2020 but had to be dramatically canceled after six sessions due to the outbreak of the COVID-19 pandemic in Spain.

Hence, a feasibility study will be conducted in autumn 2021 in some territories of the APTITUDE project. Baseline and post-intervention questionnaires will be administered to participants during a pilot intervention to redefine assessment and qualitative interviews will be carried out to check the acceptability of the intervention. Outcomes measures should be at least those included in the ICOPE instrument for the domains of cognition (Mini Metal State Examination (MMSE)), mobility (Short Physical Performance Battery (SPPB)), nutrition (Mini Nutritional Assessment (MNA)) and psychological capacity (Geriatric Depression Scale (GDS-5)). Fidelity—to which extent the intervention is delivered according to the facilitators’ guide—and adherence of participants will be assessed using an observation guideline. After feasibility and piloting, evaluation and implementation should follow to complete the four stages of the MRC framework.

## 5. Conclusions

This paper describes the development process of AMICOPE, a group-based multi-domain complex intervention built upon the ICOPE framework and aimed to improve and/or maintain IC through the promotion of physical activity, healthy nutrition, and psychological wellbeing in older people. The study is reported according to the GUIDED checklist and represents the first stage of the UK Medical Research Council framework for developing and evaluating a complex intervention.

## Figures and Tables

**Figure 1 ijerph-18-05979-f001:**
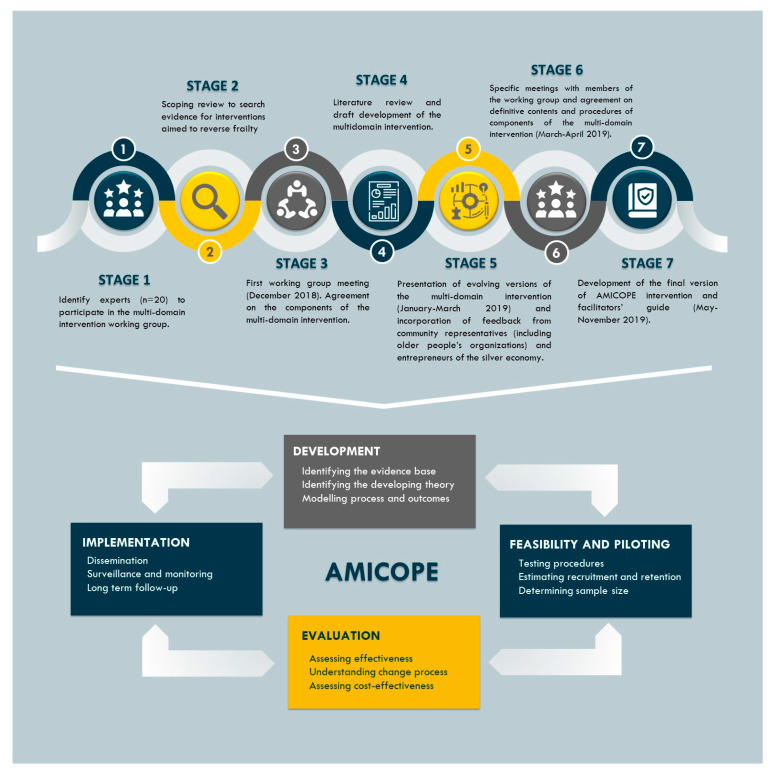
Development process of the AMICOPE intervention within the MRC framework for the development and evaluation of complex interventions. Adapted from [61] and available via license CC BY 4.0.

**Figure 2 ijerph-18-05979-f002:**
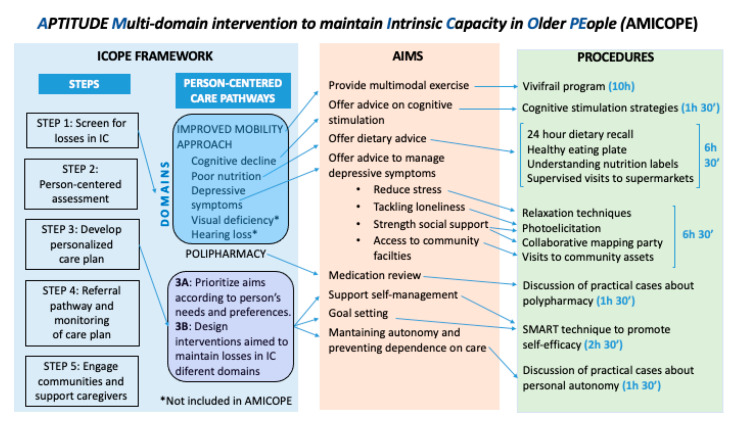
Conceptual framework of the AMICOPE multi-domain intervention.

## Data Availability

No new data were created or analyzed in this study. Data sharing is not applicable to this article.

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
