# Peer review of "A Multi-Domain Group-Based Intervention to Promote Physical Activity, Healthy Nutrition, and Psychological Wellbeing in Older People with Losses in Intrinsic Capacity: AMICOPE Development Study"

_ijerph, 2021, doi:10.3390/ijerph18115979_

Round 1

Reviewer 1 Report

Dear author, after check the second Review, some details are proposed in order to consider.Best Regards 1) Page number 1: After filiation and mail include, the author's abbreviation

2) Page 3, Line 13: the multicomponent exercise program vivifrail© is based on a series of exercises. C better in in superindex. Correct trought the text.

3) Page 5, Figure 1: stage 2 stage 5. Include names more clear, and yellow colour and black. Improve figure qualitiy.

4) Pag 5, Line 8: e vivifrail© program [63]. C in superindex

5) Page 7, Line 4,8,32: tators (version 1.0, © fundació salut i envelliment uab). Idem

6) Pag 7, Line 8: (© mikel izquierdo). I my opinion delete c with names

7) Pag 4, Line 22. Last references. Include the year in bold

Author Response

Dear reviewer,

Thank you very much for this second review of our manuscript. We have carried out a revision of English language of the manuscript.

Please find below a point-by-point response to your comments. The new revised manuscript including last changes has been included as an attachment.

1) Page number 1: After filiation and mail include, the author's abbreviation

Authors' abbreviations have been included after filiation and mail, as suggested.

2) Page 3, Line 13: the multicomponent exercise program vivifrail© is based on a series of exercises. C better in in superindex. Correct trought the text.

We have modified this according to your suggestion and changed the (C) to superindex through the text.

3) Page 5, Figure 1: stage 2 stage 5. Include names more clear, and yellow colour and black. Improve figure qualitiy.

We have changed Figure 1 to improve quality and modified the colours of stage 2 and stage 5. We think the text is clearer now. The new figure is included in the enclosed manuscript.

4) Pag 5, Line 8: e vivifrail© program [63]. C in superindex

We have modified this according to your suggestion and changed the (C) to superindex.

5) Page 7, Line 4,8,32: tators (version 1.0, © fundació salut i envelliment uab). Idem

We have modified this according to your suggestion and changed the (C) to superindex.

6) Pag 7, Line 8: (© mikel izquierdo). I my opinion delete c with names

We have deleted "(C) mikel izquierdo" according to your suggestion.

7) Pag 4, Line 22. Last references. Include the year in bold

We have revised these references and included the year in bold.

Yours sincerelly,

Sergi Blancafort

Reviewer 2 Report

Thank you for sending me this manuscript to review again. The paper is now much improved and clearer, and can be accepted with a few minor tweaks:

  • add details of intervention duration and healthcare professionals into the abstract
  • the authors mention a feasibility study being carried out 'over the next months' - in order to avoid dating the paper would state the months/seasons and year (e.g. 'summer 2021')
  • some minor English tweaks are needed (this may be addressed at proofing stage?) e.g. 'ration size' should be 'portion size' and the occasional odd grammar
  • p6 line 20 'little discussion about' - reads a bit oddly, unclear if authors meant there was a lot of consensus or that simply this wasn't discussed very much during the workshops

Author Response

Dear reviewer,

Thank you very much for this second review of our manuscript. 

Please find below a point-by-point response to your comments. The new revised manuscript including last changes has been included as an attachment.

  • add details of intervention duration and healthcare professionals into the abstract

We have added details about intervention duration and we have added information about facilitators (health and social care professionals) in the abstract. 

  • the authors mention a feasibility study being carried out 'over the next months' - in order to avoid dating the paper would state the months/seasons and year (e.g. 'summer 2021')

We have included "autumn 2021" (Pg 9, line 37) in the manuscript . This is when the feasibility study will be carried out. 

  • some minor English tweaks are needed (this may be addressed at proofing stage?) e.g. 'ration size' should be 'portion size' and the occasional odd grammar

We have added "portion size" instead of "ration size". We have also carried out extensive revision of English language of the manuscript.

  • p6 line 20 'little discussion about' - reads a bit oddly, unclear if authors meant there was a lot of consensus or that simply this wasn't discussed very much during the workshops

We have changed this sentence accordingly with your suggestion. The final sentence is:

"There was a lot of consensus on the type of intervention (group vs individual) and about who should deliver the intervention (the same facilitators for the whole intervention vs a different expert for each session)"

Yours sincerelly,

Sergi Blancafort

Reviewer 3 Report

Congratulations for your review. All is correct

Author Response

Dear reviewer,

Thank you for your comments. 

Please find enclosed a final version of the manuscript including the suggestions from other reviewers.

Yours sincerelly,

This manuscript is a resubmission of an earlier submission. The following is a list of the peer review reports and author responses from that submission.

Round 1

Reviewer 1 Report

Dear Authors:

After check the document, some changes are proposed in order to improve the firts version.

Please consider a reorganization of your article

King Rergards

Author Response

PEER REVIEW 1

  • Point 1. Section: Names; Proposal: Delete abbreviatures here
  • Response: Abbreviatures have been deleted as suggested by the reviewer.

  • Point 2. Section: Names; Proposal: Without line ([email protected])
  • Response: Underline has been removed as suggested by the reviewer.

  • Point 3. Section: Key words; Proposal: Key words are simple words
  • Response: Our initial proposal of key words has been replaced by MesH terms. The new proposal of key words is included in the manuscript.

  • Point 4. Section: Abstract; Proposal: Include the main results
  • Response: Abstract has been modified according to the suggestion of the reviewer.

  • Point 5. Section: Introduction; Proposal: Include reference

“Mobility is a critical issue for healthy ageing and preventing dependence on care. In fact, structured and adapted exercise is the preventive intervention that has shown more evidence in the management of frailty and risk of falls Underline has been removed as suggested by the reviewer.”

“Physical exercise benefits are associated with a decrease in the risk of mortality, chronic disease, institutionalization, and cognitive and functional impairment.”

  • Response: References have been included as suggested by the reviewer (page 2, linea 44-51)

  • Point 6. Section: Material and methods; Proposal: Define APTITUDE abbreviature
  • Response: The meaning of APTITUDE abbreviature has been defined and included in the manuscript (page 4, line 14)

  • Point 7. Section: Material and methods; Proposal: Define AMICOPE abbreviature
  • Response: The meaning of AMICOPE abbreviature has been defined and included in the manuscript (page 4, line 4).

  • Point 8. Section: Material and methods; Proposal: Did you considered the ethical issues?
  • Response: This CBPR was approved by its Governing Council on 30 January 2013. The updated version of the CBPR observes the recommendations of the European Charter for Researchers (Europan Commission), the European Code of Conduct for Research Integrity (ALLEA) and other documents on good scientific practice from national and international public research institutions. It was prepared and revised by the Ethics Committee on Animal and Human Experimentation (CEEAH) of the UAB and was approved by the Research Commission on 12 February 2020 and by the Governing Council on 30 September 2020.

We have included this sentence in the material and methods section (page 6, line 26):

“The study follows the Code of Good Practice in Research (CBPR) adopted by the Universitat Autònoma de Barcelona (UAB).”

  • Point 9. Section: Material and methods; Proposal: Include reference

“During the intervention development process, decisions were taken in accordance with evidence from resembling interventions delivered to frail older people, and results from our previous studies.”

  • Response: Refences have been included in the manuscript (page 4, line 41) as suggested by the reviewer.

  • Point 10. Section: Material and methods; Proposal: Include other more quality figure (figure 2)
  • Response: A new figure with higher resolution has been included in the manuscript to replace the original.

  • Point 11. Section: Material and methods; Include reference

“Evolving versions of the intervention were presented during the development process to integrate stakeholder contributions.”

  • Response: The following reference has been included in the manuscript (page 6, line 8):

Intervención multi-componente AMICOPE. Available online: https://www.aptitude-net.com/es/intervencion-multi-componente-amicope (accessed on 8 May 2021)

  • Point 12. Section: Material and methods; Proposal: Define materials, version, copyright, etc.
  • Response: A paragraph with more detailed information about the characteristics of the materials has been included as part of the description of the resulting intervention (page 7, line 4).

  • Point 13. Section: Material and methods; Proposal: Which type of strength, volume of aerobic exercises, define these concepts
  • Response: A paragraph with more detailed information about procedures for physical activity has been included in the description of the resulting intervention (page 7, line 8).

  • Point 14. Section: Results; Proposal: In this part no include references, only describe your results
  • Response: References have been removed from this section.

  • Point 15. Section: Results; Proposal: This paragraph, move to the methodological part. This is no results, is part of methods. Reorganize the document.

“The intervention consists of 12 face-to-face sessions and facilitated in groups of 8 to-12 frail older people. Sessions are held weekly for 2.5 hours during three months. Each session includes one hour of physical exercise using the Vivifrail program, and 1.5 hours dedicated to any other intervention components. The intervention will be delivered in community facilities such as senior leisure centers, civic centers, or primary care centers, and in different locations of the surroundings. Particularly, 10 of the 12 sessions will take place in a space large enough to do physical activity. For one session the whole group will move to a food store or supermarket to learn about nutritional facts. The remaining session will be devoted to visit another senior center with the purpose to know about programs and activities addressed to the community.”

  • Response: This paragraph is part of the description of the resulting intervention (AMICOPE group-based multi-domain complex intervention). As our manuscript corresponds to an intervention development study, we consider that it should remain in the section of results. We have incorporated as supplementary material the TIDIER checklist to report the full resulting intervention.
  • Point 16. Section: Results; Proposal: This paragraph, move to the methodological part. This is no results, is part of methods. Reorganize the document.

“The intervention was planned to be piloted in Catalunya and ten health and social professionals (six nurses, two physiotherapists and two physical activity trainers) were trained during November and December of 2019. A pilot of AMICOPE intervention addressed to a group of 12 frail older people started in Tortosa (Baix Ebre/Spain) in the beginning of February 2020 but had to be dramatically cancelled after six sessions due to the outbreak of the COVID-19 pandemic in Spain. A feasibility study will be conducted in the territories of the APTITUDE project to pilot the intervention, to incorporate any modifications that may improve the design, procedures, or implementation. For that purpose, one or two health or social care professionals will monitor the intervention to fill an observation log that includes several quantitative and qualitative indicators of implementation such as fidelity and adherence.”

  • Response: This paragraph describes an attempt to assess the feasibility of the intervention. Hence, it is a further step of the process of development and evaluation of complex interventions. We have slightly modified the paragraph and moved it to the end of the discussion (page 9, line 34).
  • Point 17. Section: Discussion; Proposal: In this part, include the main conclusions

“In this study we use the GUIDED checklist to describe the intervention development process for the AMICOPE group-based multi-domain intervention. We rationalize the process by detailing: the context, the aim, the recipients, the theory and evidence-base, the utilization of previous experiences, the guiding principles, the participation of stakeholders, the changes made throughout the process, and remaining uncertainties. The intervention itself is described using the TIDieR checklist and reporting of the intervention development process draws on frameworks such as the WHO Integrated Care of Older People and the UK Medical Research Council framework to develop and evaluate complex interventions.”

  • Response: This part has been modified accordingly to the suggestions of the reviewer (page 8, line 26)
  • Point 18. Section: Strengths and limitations; Proposal: This part is very long. Concrete more and NO USE REFERENCES, only describe the strengths and limitations.

“To date, the few examples found using the GUIDED checklist includes two PubMed peer-reviewed articles reporting interventions aimed to improve mental health help-seeking behaviours for male students [57], and tamoxifen adherence in breast cancer [58]; two preprints reporting interventions aimed to improve early diagnosis of cancer in primary care [59] and targeting antipsychotic prescribing to nursing home residents with dementia [60]; and one doctoral thesis reporting an arts-based intervention for patients with kidney disease [61].”

  • Response: This part has been moved before strengths and limitations in the discussion section (page 9, line 3)
  • Point 19. Section: Discussion; Proposal: Did you considered the future lines based on your results?
  • Response: A paragraph with more detailed information considering further steps of research has been included in the end of discussion (page 9, line 32).
  • Point 20. Section: Discussion; Proposal: Move this part to other paragraph NOT IN CONCLUSION

“The next step should be carrying out a feasibility study for the AMICOPE intervention, and in a later stage, assessing the effectiveness in a randomized controlled trial.”

  • Response: This part has been slightly modified and moved to the end of discussion (page 9, line 34)
  • Point 21. Section: Funding; Proposal: Move to funding paragraph

Nicolás Martínez-Velilla received funding from “la Caixa” Foundation (ID 100010434), under agreement LCF/PR/PR15/51100006.

  • Response: This part was originally in acknowledgements and has been moved to funding paragraph.
  • Point 22. Section: References; Proposal: J. Biomech (in cursive letter)
  • Response: This has been accordingly corrected in the references section.
  • Point 23. Section: References; Proposal: DOI Replicated
  • Response: This has been accordingly corrected in the references section.

Reviewer 2 Report

This manuscript reports on development of a multi-domain intervention targeted at frail older people. Overall, the paper is a good summary of how the intervention was developed. Often in frailty intervention trials there is little detail as to how interventions were developed, who was involved etc, so it is good to see this, and it is a shame the authors' pilot study had to be stopped due to Covid. The 2 diagrams are clear. 

Major points:

  • The extent of involvement of older people with frailty in intervention development is unclear. Fig1 implies only health and social care professional involvement in development, whereas the text says "community representatives and entrepreneurs of the silver economy". It is very unclear what this means and further details should be given on working group composition. Particularly, if older people were not involved in an intervention targeted at them, why was this?
  • It would be good to have a tabular summary of the salient intervention characteristics, separate to the logic model, perhaps following the TIDIER checklist. And some further intervention details, such as at what point are losses screened for? Are participants required to do 'homework' between sessions? What happens if they don't attend? Could the intervention guide be attached as an appendix if brief?
  • It's unclear where the frail older people will be recruited/referred from. Primary care? For things like medication review, do facilitators do this or refer to pharmacists?
  • Likewise the target population is "frail older people living in the community with losses in mobility, nutritional and/or psychological domains of intrinsic capacity, and without cognitive decline, visual impairment or hearing loss." As I'm sure the authors are aware, frailty can be defined and measured in many ways, and the definition used in the introduction does not easily lend itself to measurement. How will the authors define 'frail older people' to receive this intervention? How will they be screened? Likewise at what point will losses be identified? Also it seems unrealistic to try and get older participants with no visual impairment or hearing loss or MCI (esp as people may have mild levels only slightly affecting function) - are there cutoffs for this?  
  • Further detail is needed on the workshops - what decisions were brought to them, how the intervention evolved. for example, who decided how long it should be, who should deliver the intervention, group vs individual etc. Were there areas of disagreement or a lot of consensus?
  • The considerations around mobility and access need discussion if all are held in community locations. Will transport be provided for those with mobility issues?

Minor points:

  • Need a clearer rationale in intro for a group rather than individual level intervention
  • I am surprised not to see some of the most recent frailty guidelines (e.g. Dent 2019 ICFSR. J Nutr Health Aging. 2019;23(9):771-787) or multidomain systematic reviews (e.g. Apostolo 2018 JBI Database of Systematic Reviews and Implementation Reports) cited in the introduction, as these provide a good summary of current evidence of frailty interventions
  • psychological aspects only include depression. what about anxiety?
  • There are some odd phrasings to the language used throughout (e.g. abstract - "dependence, being adapted physical activity is the preventive intervention" - 'being' is not needed; or intro "reinforce caregivers (step 5)." does not make sense, do the authors mean support?; "being frail older people with the lowest income and educational level the most vulnerable" - being should be 'are' and later in the sentence); and some others throughout. Further proofing is needed. 

Author Response

PEER REVIEW 2

Major points:

  • Point 1. The extent of involvement of older people with frailty in intervention development is unclear. Fig 1 implies only health and social care professional involvement in development, whereas the text says "community representatives and entrepreneurs of the silver economy". It is very unclear what this means and further details should be given on working group composition. Particularly, if older people were not involved in an intervention targeted at them, why was this?
  • Response: Figure 1 has been accordingly modified to reflect the involvement of community representatives (including older people’s organizations) and entrepreneurs of the silver economy. Explanation is included on page 6, lines 3-13. Older people will be involved in testing procedures during the phase of feasibility and piloting, and before the evaluation (Page 9, line 44).

  • Point 2. It would be good to have a tabular summary of the salient intervention characteristics, separate to the logic model, perhaps following the TIDIER checklist. And some further intervention details, such as at what point are losses screened for? Are participants required to do 'homework' between sessions? What happens if they don't attend? Could the intervention guide be attached as an appendix if brief?

Response: A detailed table with intervention characteristics following the TIDIER checklist has been included as supplementary material.

Further intervention details regarding work at home, screening for losses and adherence are included in the text (page 8, line 8 and page 8, line 15)

The English version of the Vivifrail materials is available on line (a reference has been included). The intervention guide is not attached as an attached due to its extension (122 pages) but it will be available on-line after feasibility and piloting.

  • Point 3. It's unclear where the frail older people will be recruited/referred from. Primary care? For things like medication review, do facilitators do this or refer to pharmacists?
  • Response: Frail older people will be recruited and/or referred from primary care and community settings (page 4, line 30). With respect to medication review, a pharmacist will participate in this specific session to support facilitators and to provide specific counseling to older people (page 7, line 29).

  • Point 4. Likewise the target population is "frail older people living in the community with losses in mobility, nutritional and/or psychological domains of intrinsic capacity, and without cognitive decline, visual impairment or hearing loss." As I'm sure the authors are aware, frailty can be defined and measured in many ways, and the definition used in the introduction does not easily lend itself to measurement. How will the authors define 'frail older people' to receive this intervention? How will they be screened? Likewise at what point will losses be identified? Also it seems unrealistic to try and get older participants with no visual impairment or hearing loss or MCI (esp as people may have mild levels only slightly affecting function) - are there cutoffs for this?  
  • Response: We have changed slightly the definition of frailty accordingly to this suggestion (page 2, line 10).

Regarding the target population of AMICOPE, our study focuses more on the concept of         intrinsic capacity developed by the WHO rather than frailty. We go ahead the phase of frailty            and we analyze a phase with lower prevalence of diseases and disability, visual impairment          and hearing loss. We have included a short sentence in the manuscript about this relevant    issue pointed by the reviewer (page 4, line 32).

  • Point 5. Further detail is needed on the workshops - what decisions were brought to them, how the intervention evolved. for example, who decided how long it should be, who should deliver the intervention, group vs individual etc. Were there areas of disagreement or a lot of consensus?
  • Response: Further details about the workshops has been incorporated in the text (page 6, line 13-22)

  • Point 6. The considerations around mobility and access need discussion if all are held in community locations. Will transport be provided for those with mobility issues?
  • Response: Unfortunately transport are not provided for those with mobility issues. So, participants who are not able to go the intervention site by their own, will be excluded. This information has been included in the text (page 4, line 31)

Minor points:

  • Point 7. Need a clearer rationale in intro for a group rather than individual level intervention
  • Response: A paragraph has been included in the introduction with a clearer rationale for group-based interventions rather than individual (page 3, line 42)

  • Point 8. I am surprised not to see some of the most recent frailty guidelines (e.g. Dent 2019 ICFSR. J Nutr Health Aging. 2019;23(9):771-787) or multidomain systematic reviews (e.g. Apostolo 2018 JBI Database of Systematic Reviews and Implementation Reports) cited in the introduction, as these provide a good summary of current evidence of frailty interventions
  • Response: Thank you very much for your suggestions. We have included these references in the introduction (Page 2, line 46 and 49).

  • Point 9. psychological aspects only include depression. what about anxiety?
  • Response: We have included anxiety within psychological aspects in some parts of the manuscript (page 2, line 38; page 3, line 4 and 31). Indeed, and according to ICOPE strategy, depressive symptoms are an important component of psychological capacity, but only one dimension. There are other aspects (anxiety is specifically mentioned) that need to be addressed.

  • Point 10. There are some odd phrasings to the language used throughout (e.g. abstract - "dependence, being adapted physical activity is the preventive intervention" - 'being' is not needed; or intro "reinforce caregivers (step 5)." does not make sense, do the authors mean support?; "being frail older people with the lowest income and educational level the most vulnerable" - being should be 'are' and later in the sentence); and some others throughout. Further proofing is needed. 
  • Response: Thank you very much for your suggestion. The updated versions of the manuscript has been accordingly read and revised by a native English-speaking colleague.

Reviewer 3 Report

I want to say you, that this manuscript it's very good, but i add a major comments to improve substantially.

INTRODUCTION: 

The intoduction its correct, but you should add information about different exercise programs to improve fall risk, intensive capacity and similar.

METHODS: Why did you use GUIDED checklist. You say that one problem it's that in Pubmed there are not manuscript with GUIDED checklist, but there are manuscript with CERT checklist. You should explain this part as a limitation of this study.

Multicomponent Exercise Program (MEP), it's a new method to improve health in general in older people, but you should explain how control intensity, volume, etc.. and if you use strength or resistance, aerobic and coordination and if you add cognitive exercise in this model.

Other problem is COVID, you should add modification in your intervention, if this was a Clinical Trial, you would have lost all information. Can you train during pandemia? If is correct, how did you adapt the exercise?

DISCUSION: This part it's correct, but if you add information about CERT checklist, you should add information in other intervention controlled with CERT.

Author Response

PEER REVIEW 3: 

Comments and Suggestions for Authors 

I want to say you, that this manuscript it's very good, but i add a major comments to improve substantially. 

INTRODUCTION:  

  • Point 1. The intoduction its correct, but you should add information about different exercise programs to improve fall risk, intensive capacity and similar.
  • Response: A paragraph with information about different exercise programs has been added in the introduction of the manuscript (page 2, line 53 until page 3, line 4).

  

METHODS:

  • Point 2. Why did you use GUIDED checklist. You say that one problem it's that in Pubmed there are not manuscript with GUIDED checklist, but there are manuscript with CERT checklist. You should explain this part as a limitation of this study.
  • Response: A paragraph has been added in the manuscript explaining why we used the GUIDED checklist (page 4, line 5). The lack of availability of examples in the literature using the GUIDED checklist has been included in the manuscript as a limitation (page 9, line 28).

  • Point 3. Multicomponent Exercise Program (MEP), it's a new method to improve health in general in older people, but you should explain how control intensity, volume, etc.. and if you use strength or resistance, aerobic and coordination and if you add cognitive exercise in this model.

  • Response: Additional information about the multicomponent exercise program (how to control volume and intensity; use of strength, resistance, aerobic and coordination) has been incorporated in the manuscript. (page 7, line 10)

  • Point 4. Other problem is COVID, you should add modification in your intervention, if this was a Clinical Trial, you would have lost all information. Can you train during pandemia? If is correct, how did you adapt the exercise?

  • Response: This study describes a group-based intervention with a strong social support component. Unfortunately, training of facilitators and delivery of the intervention to frail older people has not been able during the pandemic. The physical activity program Vivifrail has been recorded in a series of videos which are available on line and where it is explained how to perform each of the exercises.

DISCUSION:

  • Point 5. This part it's correct, but if you add information about CERT checklist, you should add information in other intervention controlled with CERT.

  • Response: Information about other interventions reported by CERT has been added in the discussion of the manuscript. (page 9, line 10) .